# Single-Site Sutureless Partial Nephrectomy for Small Exophytic Renal Tumors

**DOI:** 10.3390/jcm9113658

**Published:** 2020-11-13

**Authors:** Ching-Chia Li, Tsu-Ming Chien, Shu-Pin Huang, Hsin-Chih Yeh, Hsiang-Ying Lee, Hung-Lung Ke, Sheng-Chen Wen, Wei-Che Chang, Yung-Shun Juan, Yii-Her Chou, Wen-Jeng Wu

**Affiliations:** 1Department of Urology, Kaohsiung Medical University Hospital, Kaohsiung 80756, Taiwan; ccli1010@hotmail.com (C.-C.L.); shpihu73@gmail.com (S.-P.H.); hunglungke@yahoo.com.tw (H.-L.K.); CARL0815@gmail.com (S.-C.W.); u96000018@kmu.edu.tw (W.-C.C.); juanuro@gmail.com (Y.-S.J.); yihech@kmu.edu.tw (Y.-H.C.); 2Department of Urology, Faculty of Medicine, College of Medicine, Kaohsiung Medical University, Kaohsiung 80756, Taiwan; 3Graduate Institute of Clinical Medicine, College of Medicine, Kaohsiung Medical University, Kaohsiung 80756, Taiwan; 4Department of Urology, Kaohsiung Municipal Ta-Tung Hospital, Kaohsiung 80145, Taiwan; patrick1201.tw@yahoo.com.tw (H.-C.Y.); ashum1009@hotmail.com (H.-Y.L.)

**Keywords:** partial nephrectomy, single site surgery, sutureless

## Abstract

Partial nephrectomy (PN) is the standard procedure for most patients with localized renal cancer. Laparoscopy has become the preferred surgical approach to target this cancer, but the steep learning curve with laparoscopic PN (LPN) remains a concern. In LPN intracorporeal suturing, the operation time is further extended even under robot assistance, a step which prolongs warm ischemic time. Herein, we shared our experience to reduce the warm ischemia time, which allows surgeons to perform LPN more easily by using a combination of hemostatic agents to safely control parenchymal bleeding. Between 2015 and 2018, we enrolled 52 patients who underwent LPN in our hospital. Single-site sutureless LPN and traditional suture methods were performed in 33 and 19 patients, respectively. Preoperative, intra-operative, and postoperative variables were recorded. Renal function was evaluated by estimated glomerular filtration rate (eGFR) pre- and postoperatively. The average warm ischemia time (sutureless vs. suture group; 11.8 ± 3.9 vs. 21.2 ± 7.2 min, *p* < 0.001) and the operation time (167.9 ± 37.5 vs. 193.7 ± 42.5 min, *p* = 0.035) were significantly shorter in the sutureless group. In the sutureless group, only 2 patients suffered from massive urinary leakage (>200 mL/day) from the Jackson Pratt drainage tube, but the leakage spontaneously decreased within 7 days after surgery. eGFR and serum hemoglobin were not found to be significantly different pre- and postoperatively. All tumors were removed without a positive surgical margin. All patients were alive without recurrent tumors at mean postoperative follow-ups of 29.3 ± 12.2 months. Single-site sutureless LPN is a feasible surgical method for most patients with small exophytic renal cancer with excellent cosmetic results without affecting oncological results.

## 1. Introduction

In 2009, the American Urological Association (AUA) [1] recommended partial nephrectomy (PN) as the reference standard treatment for most clinical T1 renal masses, even in individuals with a normal contralateral kidney, due to its similar efficacy to radical nephrectomy while also preserving kidney tissue. Since that time, a review of nephrectomy records submitted as part of the American Board of Urology surgeon certification/recertification process revealed that the use of PN has increased from 25% to 39% in all nephrectomies [2]. PN preserves kidney function better and limits long-term development of metabolic and cardiovascular disorders. The European Association of Urology has also considered PN the treatment of choice for T1b renal cell carcinoma (RCC) [3].

Open PN remains the gold standard procedure in most patients with localized renal cancer. Though no randomized controlled studies have compared the safety and oncological outcomes in terms of renal function and surgical margins, the steep learning curve with laparoscopic partial nephrectomy (LPN) remains a concern [4]. LPN is a technically demanding procedure, even under robotic assistance. Several important challenges, such as preventing perioperative bleeding, reaching hyperthermia after renal artery clamping, reducing warm ischemia time, and performing laparoscopic intracorporeal suturing, must be met during the operation. Despite the ability to achieve renal hyperthermia by delivering cold saline into the renal pelvis, the cooling effect is not qualified during laparoscopic surgery. Gill et al. [5] reported a novel method using ice slush around the kidney; however, this is difficult to replicate during the laparoscopic procedure. Because it is difficult to achieve renal hypothermia during LPN, it is important to reduce the warm ischemia time, which is understood to correlate with subsequent return of renal function [6]. Traditional clamping procedures require a significant warm ischemia time during the suturing process. Hemostatic suturing plays a vitally important role, even in the current era of early unclamping [7], selective clamping [8], and unclamping techniques [9,10,11]. With the introduction of hemostatic agents and improvements in surgical equipment allows for the resection of renal tumors without intracorporal suturing [12,13,14,15,16]. The suture method might also have contributed to the occurrence of pseudoaneurysms after the closure of renal defects [17]. Recently, there has been a growing application in laparoscopic single-site surgery that uses a single skin incision to gain access to the target operation site [15,16]. Single-site approach tries to minimize the rare port-related complications and fasten the postoperative recovery with excellent cosmetic results [15,16]. Robotic-assisted surgery is the new gold standard for uro-oncological surgery. However, the rigid instrumentation and the need for adaptation to the existing platform make the widespread use of these single- site surgeries difficult.

We previously shared our “pressure-cooker” method of performing LPN without intracorporeal suturing [12]. In the current study, we present our technique of single-site sutureless LPN. Our method is shown to reduce the warm ischemia time, and we believe that this technique allows surgeons to perform LPM more easily and effectively with fewer complications for those who lack experience in intracorporeal suturing.

## 2. Materials and Methods

### 2.1. Patient

In total, 116 consecutive patients with a renal tumor between 2015 and 2018 were sampled at the Kaohsiung Medical University Hospital in Kaohsiung, Taiwan. We firstly excluded metastatic tumors (*N* = 29). Patients with T2 renal tumor were also excluded (*N* = 31). Moreover, we excluded the two follow-up patients we lost, as well as the patient with a bilateral tumor. A total of 52 patients underwent LPN and were included in the current study. Single-site sutureless LPN and traditional suture methods were performed in 33 and 19 patients, respectively. All patients were informed of the potential complications and risks of the novel techniques. The study was conducted according to the principles of the Declaration of Helsinki and supervised by the local Ethics Committee of the Kaohsiung Medical University Hospital (KMUHIRB-E(I)-20180174). Written informed consent was obtained from all patients prior to their enrollment in the study. Patients with localized renal parenchymal tumor (stage T1N0M0) without endophytic properties or tumor located <4 mm from the collecting system were included. We excluded patients with suspected lymph node or distant metastasis. We quantified the anatomical characteristics of the renal masses using the RENAL nephrometry score [18]. In total, 52 patients who underwent LPN were enrolled in the study and had at least a one year follow-up (Figure 1). The authors confirmed that all ongoing and related trials for this intervention were registered.

### 2.2. Approach

We previously published an article that reported our basic sutureless LPN method [12]. Patients were placed in flank position with the lesion site elevated to 90 degrees. The surgeon and assistant stood facing the patient’s back. The length of the skin incision was approximately 2.5–3.5 cm according to the tumor diameter. The port incision was made just below the 12th rib in the posterior axillary line. All procedures were performed using the retroperitoneal approach. A balloon dilator was used to create the retroperitoneal space, which was entered via the exposed thoracolumbar fascia, irrespective of their location. We used the LagiPort (Lagis, Inc., Taichung, Taiwan), a multi-instrument access port designed especially for single-site LPN (Figure 2). Gerota’s fascia was dissected anteriorly and posteriorly. Next, an incision was made to mobilize the kidney from the perirenal fat, revealing the renal artery and primary tumor. If the tumor margin was not clear, intraoperative ultrasonography was used to better visualize the tumor margin. A fat pad from the perirenal space was prepared and was located as far away from the tumor as possible. 

### 2.3. Tumor Excision: The “Pressure Cooker” Method

In the selective renal artery non-clamping patients, a harmonic scalpel was used to remove the tumor, leaving a 0.5 to 1 cm safety margin. In the renal clamping group, the tumor was excised using laparoscopic scissors with bulldog clamps. Vascular disruption with excision was extensively fulgurated. For this procedure, we used monopolar coagulation via laparoscopic scissors to seal off the cross-section of renal calyx or pelvis if any collecting system disruptions are noted. After tumor removal, a hemostatic matrix (FloSeal; Baxter Healthcare, Zurich, Switzerland) was placed into the renal cavity, and a fibrin sealant (Tisseel; Baxter) was injected to cover the entire hemostatic matrix and the surrounding normal renal tissue. At the end of the surgery, the fat pad was placed to cover all areas coated with fibrin sealant, and the bulldog clamp was detached. The fat pad covering should be accomplished within 20 s to prevent solidifying of the fibrin sealant. The fat pad adhered to the periphery of the incision field, and the hemostatic matrix was “cooked” and closed off underneath. After the gelatin matrix and thrombin component were combined, the hemostatic matrix expanded around 20% of the volume upon contact with blood or urine. This reaction occurred soon after removing the bulldog clamp. The hemostatic matrix was engorged within the airtight space covered by the fat pad just like a “pressure cooker,” causing extra external pressure to compress the postoperative bleeding (Figure 3). The tumor specimen was removed directly through the port using a laparoscopic grasper. We routinely placed a drainage tube after the surgery.

### 2.4. Statistical Methods

All values are expressed as a mean ± standard deviation. Differences between categorical parameters were assessed using a χ^2^ or Fisher’s exact test, as appropriate. A Fisher’s exact test was used when the sample number was small. Continuous parameters were assessed by using a *t*-test or Mann–Whitney–Wilcoxon test. The threshold for statistical significance was set at *p* < 0.05. SPSS 20.0J (SPSS Inc., Chicago, IL, USA) and used for all statistical analyses.

## 3. Results

### 3.1. Study Population 

The preoperative data are shown in Table 1. The average patient age was older in the sutureless group. Twenty-four patients (46.1%) were female. The patient population was generally non-obese with a mean body mass index of 26.8 ± 3.3 (range: 21.9–38.1). Preoperative American Society of Anesthesiologists and Eastern Cooperative Oncology Group scores were 1.2 ± 0.4 (range: 1.0–2.0) and 0.3 ± 0.4 (range: 0–1), respectively. Twenty-nine patients had a left-sided renal mass. The average tumor size was 2.6 ± 1.1 cm (range: 1.5–5.0 cm). The mean R.E.N.A.L. nephrometry score [18] was 5.8 ± 1.5 (range: 4.0–9.0). 

### 3.2. Surgical Outcomes

The average operation time was 177.3 ± 40.9 min (range: 100–250 min). To achieve renal hilar control, the clampless method was used in 7 patients due to tumors in exophytic locations or the majority of tumors had a distinct fibrotic capsule. Bulldog clamps were used for temporary renal artery occlusion in the remaining 27 patients. The average warm ischemia time was 15.5 ± 7.1 min (range: 8–26 min). The renal clamping strategy was made according to the surgeon, preoperative imaging, intraoperative findings, and intraoperative ultrasound. Mean estimated blood loss was 102.4 ± 97.2 mL (range: 10.0–430.0 mL). Only 3 patients required a perioperative blood transfusion due to large tumor burden. Conversion to conventional laparoscopy or open surgery was not necessary (Table 2). We did not perform the renal cooling technique. After the operation, the renal tumor was removed from the single-site wound. In total, 5 patients had obvious collecting system disruption during the procedures. We did not perform reconstruction of the collecting system. Only 2 patients suffered from massive urinary leakage (>200 mL/day) from the Jackson Pratt drainage tube (Table 3), but the leakage spontaneously decreased within 7 days after the surgery without requiring additional surgery. The mean length of hospital stay was 5.6 ± 1.3 days. The average warm ischemia time (sutureless vs. suture group; 11.8 ± 3.9 vs. 21.2 ± 7.2 min, *p* < 0.001) and the operation time (167.9 ± 37.5 vs. 193.7 ± 42.5 min, *p* = 0.035) were significantly shorter in the sutureless group.

### 3.3. Histopathological Outcome

The pathological results revealed clear cell RCC in 28 patients (53.8%; pT1a in 22 and pT1b in 6), angiomyolipoma in 10 (19.2%), oncocytoma in 5 (9.6%), papillary RCC in 5 (9.6%; all pT1a), and chromophobe RCC in 1 (1.9%; pT1a) (Table 3). One oncocytoma and one angiomyolipoma patient with positive surgical margins received a close follow-up ultrasound and computed tomography scans. Neither the residual tumor nor recurrence were observed in an imaging study after a 36 month follow-up. All patients were alive without recurrent tumors at a mean postoperative follow-up of 29.3 ± 12.2 months (range: 12.0–46.0 months).

### 3.4. Renal Function and Hemoglobin Level

The preoperative and postoperative estimated glomerular filtration rate (eGFR) was 79.7 ± 21.1 and 70.3 ± 25.2, respectively. There was no significant decrease in eGFR level (*p* = 0.592). A mild decrease in hemoglobin level was observed (preoperative vs postoperative; 13.9 ± 1.4 vs 13.4 ± 1.4; *p* = 0.04) (Table 2 and Table 3). Notably, the average skin incision was 2.8 ± 1.2 cm with excellent cosmetic outcomes.

## 4. Discussion

PN was initially reported in 1993, wherein McDougall et al. [19] first reported a wedge resection technique for the removal of small, low-stage renal masses via LPN. Since then, LPN has been increasingly used due to refined laparoscopic suturing techniques and the availability of hemosealant substances. Although no randomized study has compared safety and oncological outcomes between LPN and the open technique, the main concern with LPN has always been the steep learning curve [4]. Stifelman el al. [20] reported the first robotic-assisted (RA) PN in 2005, demonstrating that this approach allowed for accurate lesion resection and easier reconstruction of the renal defect. A recent U.S. study [21] using the Nationwide Inpatient Sample database determined practice patterns and perioperative outcomes of open and minimally invasive PN, revealing that RAPN is currently performed more commonly than is LPN. Conversely, LPN is more widely used (69.8%) in minimally invasive procedures compared to RAPN (30.2%) in the U.K [22]. A recent meta-analysis [23] combining 4919 patients from 25 studies (RAPN in 2681 and LPN in 2238) revealed no significant differences between the 2 groups in terms of age, sex, laterality, and final malignant pathology; however, the tumor was larger, with higher mean R.E.N.A.L. nephrometry scores in the former group. Patients treated with RAPN had a decreased likelihood of conversion to open surgery compared to those treated with LPN. RAPN also was associated with reduced complications, fewer positive margins, and shorter warm ischemia time [23]. Potential disadvantages of RAPN included cost, training, setup time, and lack of tactile sensation or haptics. The robotic procedure had lower odds of advantages compared to LPN, except for hospital charges. Nonetheless, LPN still has a competitive value in patients with small exophytic renal tumors. The major concern with LPN is the learning curve. Our technique provides a feasible method without the use of intracorporeal suturing and achieves excellent functional outcomes without affecting oncological results. At our institution, we started performing LPN in 2003 and single-site LPN in 2013. We have also performed RAPN for large renal tumors since 2015. In recent years, single-site LPN has been our standard operation for patients with small renal masses. For those with larger tumors, open and RAPN are two of our most utilized surgical procedures. 

Our study identified 5 patients with obvious disruption of the collecting system. We did not perform traditional suture repair of the collecting system. Ploussard et al. [24] showed that even after deep one-third PN, the combinations of FloSeal and Tisseel appeared to sufficiently control the major medullary vascular injuries and replace the conventional deep medullary sutures without compromising operative outcomes in a pig model. We previously described our methods using combinations of hemostatic agents with a fat pad around the outer layer of the kidney. The fat pad encapsulated the hemostatic agents within the tumor-excised cavity, supplementing structural support of the expanding and swelling action of FloSeal after it interacts with blood or urine from within. The extra external pressure provided by the fat pad acts in theory like a “pressure cooker” in preventing postoperative bleeding. The suture procedure may occlude unnecessary vessels at the suture site, leading to areas of kidney necrosis in the region. By decreasing the risk of unnecessary segmental vessel occlusion, the potential advantages may be noted during functional and vascular follow-up examinations.

Pathologic difference is an important prognostic factor for renal tumor [25]. Exophytic renal tumors tended to be associated with lower pathologic grade and the presence of papillary renal cell carcinoma subtype when compared with endophytic renal tumors [25]. Papillary renal cell carcinoma is reported to have better outcomes than clear cell renal cell carcinoma in patients without metastases [26]. Furthermore, the presence of an angular interface with the normal renal parenchyma is strongly related to benignity in an exophytic renal mass. Thus, a simple assessment of the angular interface sign can be considered as an additional parameter to characterize exophytic renal masses [27]. Optimal follow-up or therapy for patients with renal tumors should be assigned according to the tumor stage and subtype. The aforementioned information may be useful when small tumors are being considered for watchful waiting or ablative therapies. 

The most important factor in preserving renal function during PN is the percent of nephron mass preserved [6,28,29,30]. In our series, one of our main findings relates to nephron mass preservation, which is of primary importance for functional recovery, consistent with reports from other studies that eGFR of small renal cancer was not significantly different pre- and postoperatively [10,11]. Traditionally, LPN relies on clamping the main artery, with ischemia time considered to correlate with postoperative renal function. Gill et al. [5] shared a novel technique of laparoscopic renal hypothermia with intracorporeal ice slush during LPN. However, this cooling procedure was not easy to replicate during laparoscopic surgery; therefore, it is important to reduce the warm ischemia time. A threshold may exist after the damage from ischemia begins. Thompson et al. [6] demonstrated that every minute is important, and 25 min was considered a safe threshold in patients with a solitary kidney. Lane et al. [30] evaluated early and late renal functional outcomes in 1132 patients with 2 functioning kidneys, showing that a warm ischemia time of <20 min is not associated with clinically relevant functional loss compared to that of alternative techniques. Gill et al. [9] was the first to describe a technique of “zero ischemia,” which focused special attention on selective branch microdissection of renal vessels in the renal sinus; transient, pharmacologically induced blood pressure reduction timed to coincide precisely with excision of the deep part of the tumor; laparoscopic ultrasound to score the proposed resection margin; and clip ligation of any specific tertiary or quaternary renal artery branches supplying the tumor. The effort to minimize ischemia is accompanied by increased blood loss during the procedure. The potential impact on the surgical margin may be influenced by the lack of a clear operative field, which may bring surgical challenges for inexperienced operators, especially in larger renal tumors [31]. A current review paper [31] argued that newer strategies focusing on selective clamping and non-clamping can make a complex surgery even more challenging, which may serve to limit the widespread use of LPN for management of renal cancers. We believe that our technique should be used in single-site sutureless LPN to improve not only the warm ischemia time but also allows surgeons to perform LPM more easily and more effectively. 

Our study has several limitations. First, this was not a randomized prospective analysis and was composed of a relatively small cohort. An important selection bias might have resulted in satisfied surgical outcomes due to all participants were patients with exophytic renal tumors. The use of this technique for endophytic tumor still needs to be explored. Our method allows surgeons to perform LPN more easily and effectively with fewer complications compared to the open method.

## 5. Conclusions

In conclusion, single-site sutureless LPN is a feasible surgical method for most patients with small exophytic renal cancer with excellent cosmetic results without affecting oncological results. Further prospective studies with longer follow-up are needed to observe the oncological safety of the technique.

## Figures and Tables

**Figure 1 jcm-09-03658-f001:**
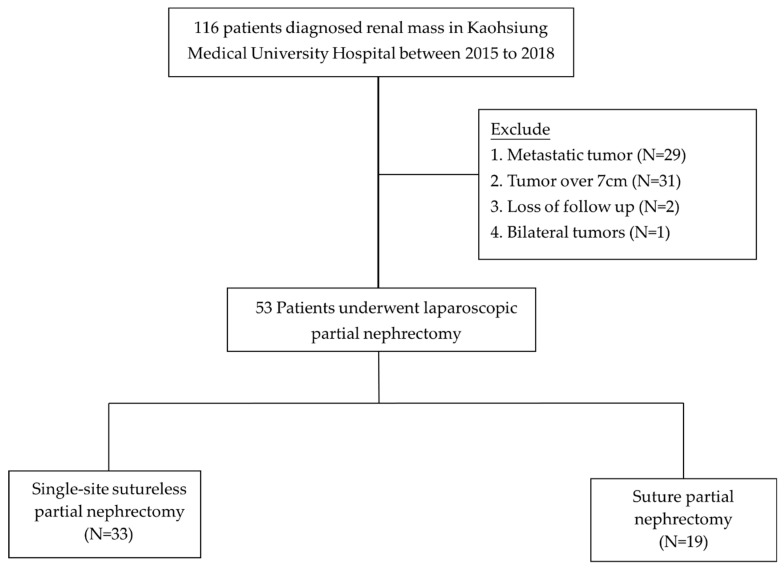
Patient enrollment for patients with renal tumor underwent surgical interventions.

**Figure 2 jcm-09-03658-f002:**
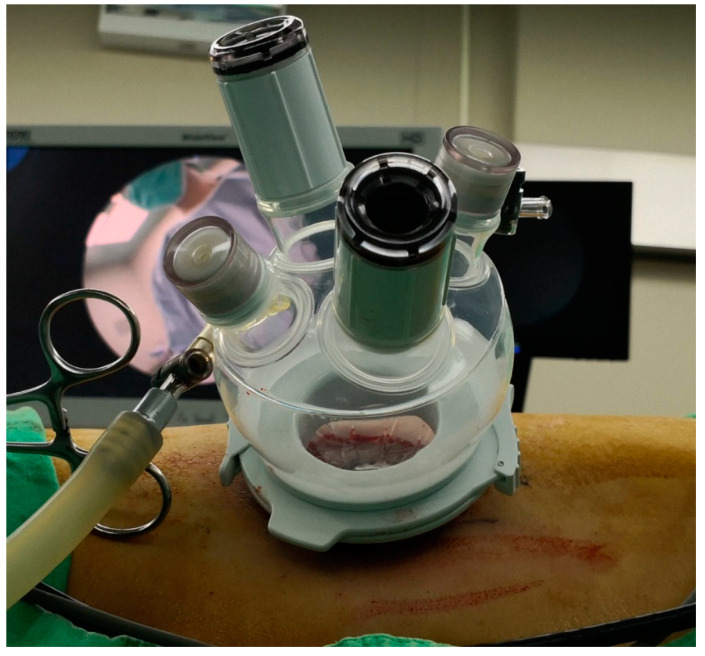
Placement of the LagiPort trocar.

**Figure 3 jcm-09-03658-f003:**
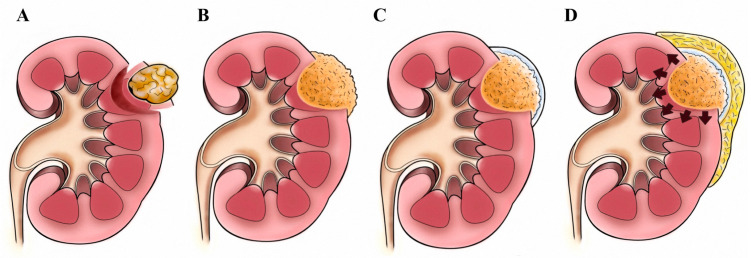
(**A**) A defect after tumor was removed. (**B**) FloSeal was placed into the defect of the kidney. (**C**) Tisseel was then injected to cover the whole hemostatic matrix and surrounding normal kidney surface. (**D**) A fat pad was placed on the top the field covered with Tisseel. FloSeal will swell in the airtight space, like a “pressure cooker”.

**Table 1 jcm-09-03658-t001:** Preoperative data on patients who underwent surgery.

Preoperative Variable	Total (*N* = 52)	Sutureless Group (*N* = 33)	Suture Group (*N* = 19)	*p* Value
Age (Mean ± SD), years	57.1 ± 10.7	59.7 ± 11.1	52.5 ± 8.5	0.013
Gender (female/male ratio)	0.46	0.48	0.42	0.715
BMI (Mean ± SD), kg/m^2^	26.8 ± 3.3	26.8 ± 3.2	26.7 ± 3.6	0.917
Left/right kidney	29/23	18/15	11/9	0.974
ASA score (Mean ± SD)	1.2 ± 0.4	1.2 ± 0.4	1.3 ± 0.5	0.366
ECOG score (Mean ± SD)	0.3 ± 0.4	0.3 ± 0.5	0.3 ± 0.4	0.751
Tumor size (Mean ± SD), cm	2.6 ± 1.1	2.7 ± 1.1	2.5 ± 1.0	0.538
R.E.N.A.L. score (Mean ± SD)	5.8 ± 1.5	5.7 ± 1.5	5.9 ± 1.7	0.626
Preoperative eGFR, mL/min/m^2^	79.7 ± 21.1	76.6 ± 22.4	85.1 ± 18.1	0.146
Preoperative hemoglobin, g/dL	13.9 ± 1.4	13.9 ± 1.3	14.0 ± 1.5	0.884

**Table 2 jcm-09-03658-t002:** Intraoperative and postoperative data on patients who underwent surgery.

Intra-Operative and Postoperative Variable	Total (*N* = 52)	Sutureless Group (*N* = 33)	Suture Group (*N* = 19)	*p* Value
Operation time (Mean ± SD), min	177.3 ± 40.9	167.9 ± 37.5	193.7 ± 42.5	0.035
Renal artery control (clamped)	45 (86.5%)	27 (81.8%)	18 (94.7%)	0.189
Warm ischemia time (Mean ± SD), min	15.5 ± 7.1	11.8 ± 3.9	21.2 ± 7.2	<0.001
Blood loss (Mean ± SD), mL	102.4 ± 97.2	104.0 ± 105.8	99.7 ± 83.6	0.881
Transfusion	3 (5.8%)	1 (3.0%)	2 (10.5%)	0.264
Conversion to conventional laparoscopy	0	0	0	
Hospital stay (Mean ± SD), day	5.6 ± 1.3	5.6 ± 1.5	5.5 ± 1.6	0.848
Postoperative eGFR, mL/min/m^2^	70.3 ± 25.2	69.6 ± 24.3	72.2 ± 21.8	0.340
Postoperative hemoglobin, g/dL	13.4 ± 1.4	13.3 ± 1.3	13.5 ± 1.5	0.642
Skin incision (Mean ± SD), cm	2.8 ± 1.2	2.8 ± 1.1	2.9 ± 1.4	0.771

**Table 3 jcm-09-03658-t003:** Histopathological and follow-up results on patients who underwent surgery.

Histopathological Variable	Total (*N* = 52)	Sutureless Group (*N* = 33)	Suture Group (*N* = 19)
Clear cell RCC			
pT1a	22 (42.3%)	14 (42.4%)	8 (42.1%)
pT1b	6 (11.5%)	4 (12.1%)	2 (10.5%)
Papillary RCC			
pT1a	5 (9.6%)	3 (9.1%)	2 (10.5%)
Chromophobe RCC			
pT1a	1 (1.9%)	1 (3.0%)	0 (0%)
Angiomyolipoma	10 (19.2%)	8 (24.2%)	2 (10.5%)
Oncocytoma	5 (9.6%)	3 (9.1%)	2 (10.5%)
Complications			
Prolong urine leakage	2 (3.8%)	2 (6.1%)	0 (0%)
Positive surgical margin	2 (3.8%)	2 (6.1%)	0 (0%)
Cancer recurrence	0 (0%)	0 (0%)	0 (0%)
Duration of follow-up (Mean ± SD), months	29.3 ± 12.2	27.5 ± 10.4	35.2 ± 14.3

RCC: Renal cell carcinoma.

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
