# Peer review of "Single-Site Sutureless Partial Nephrectomy for Small Exophytic Renal Tumors"

_jcm, 2020, doi:10.3390/jcm9113658_

Round 1
Reviewer 1 Report
Dear Authors,
I read with interest your manuscript titled: "Single-site Sutureless Partial Nephrectomy for Small 2 Renal tumor ". The topic is interesting. However, nowadays robot-assisted surgery is the new gold standard for uro-oncological surgery and I am not sure that the single site sutureless LPN technique could be of great interest for the readers. Here my comments.
Abstract: The aim of the study is not clear. In the first lines the Authors generally stated that “LPN learning curve remains a concern. Here we shared our experience to reduce the warm ischemia time and shorten the learning curve in performing LPN.”, however, all the manuscript is about the single site sutureless PN. Please be more specific.
Introduction:
Please enrich the introduction with some references on single site sutureless PNL, which is the main topic of your manuscript
“We believe this technique will shorten the learning curve in performing LPN for surgeons who lack experience in intracorporeal suturing.” this should be the primary endpoint. In order to demonstrate that the Authors should show learning curve analyses for both standard LPN and single site sutureless LPN according to the surgical experience. To demonstrate comparable results in terms of perioperative and postoperative between the two different technique do not mean that single site sutureless LPN learning curve will be shorter than the standard one.
Methods: The Authors stated “Patients with localized renal parenchymal tumor (stage T1N0M0) without endophytic properties or tumor located <4 mm from the collecting system were included. We excluded patients with suspected lymph node or distant metastasis”. This is an important selection bias. The Authors should use this technique on different kind of small renal masses, not only the exophytic ones.
Statistical analyses and Table 3: There is no clue of Kaplan Meyer methodology. Moreover, the legend of Table 3 stated :”cancer specific survival for the two groups” However, I do not see any curves and it is not clear how the Authors estimated the cancer specific mortality and at which time point. Similar problem for the recurrence free survival.
Discussion and conclusions: Discussion is confusing. Here, the Authors refers to several studies which are not about the main point of this manuscript.
Moreover, the Authors stated:” We believe that our 265 technique should be used in single-site sutureless LPN to improve not only the warm ischemia 266 time but also the learning curve of surgeons. This technique can also be used in traditional open 267 PN, LPN, and even RAPN for surgeons lacking experience in intracorporeal suturing.” With the results presented these conclusions are too strong. The Authors did not report or use the technique presented in this manuscript for open or robot assisted surgery.
Reviewer 2 Report
Authors present a series of single site sutureless PN compared to standard LPN.
The paper could benefit from expanded discussion, characterization, and examples of the exophytic characterisitics of the tumor.
The single site technique could be discussed further as to limitations and challenges with this approach based on tumor location or other issues that may benefit readers.
An explanation of the superior GFR of the sutured LPN category should be provided. This does not seem consistent with much of the discussion. Perhaps an error as the values were higher post-operatively.
